# Minimal Clinically Important Difference (MCID) in the Functional Status Measures in Patients with Stroke: Inverse Probability Treatment Weighting

**DOI:** 10.3390/jcm12185828

**Published:** 2023-09-07

**Authors:** Yu-Chien Chang, Hsiu-Fen Lin, Yu-Fu Chen, Hong-Yaw Chen, Yu-Tsz Shiu, Hon-Yi Shi

**Affiliations:** 1Division of Neurology, Department of Internal Medicine, Yuan’s General Hospital, Kaohsiung 80249, Taiwan; yu8701054@yahoo.com.tw; 2Department of Neurology, Kaohsiung Medical University Hospital, Kaohsiung 80756, Taiwan; flin@kmu.edu.tw; 3Department of Neurology, Kaohsiung Medical University, Kaohsiung 80708, Taiwan; 4Department of Clinical Education & Research, Yuan’s General Hospital, Kaohsiung 80249, Taiwan; y3903@yuanhosp.com.tw; 5Superintendent and Division of Digestive Surgery, Department of Surgery, Yuan’s General Hospital, Kaohsiung 80249, Taiwan; chy_med@yuanhosp.com.tw; 6Department of Healthcare Administration and Medical Informatics, Kaohsiung Medical University, Kaohsiung 80708, Taiwan; u109572006@gap.kmu.edu.tw; 7Department of Business Management, National Sun Yat-sen University, Kaohsiung 80424, Taiwan; 8Department of Medical Research, Kaohsiung Medical University Hospital, Kaohsiung 80756, Taiwan; 9Department of Medical Research, China Medical University Hospital, China Medical University, Taichung 40402, Taiwan

**Keywords:** ischemic stroke, hemorrhagic stroke, post-acute care, minimal clinically important difference, deterioration

## Abstract

This study proposed to evaluate the temporal trend, define the minimal clinically important difference (MCID) for five functional status measures, and identify risk factors for reaching deterioration in the MCID. This prospective cohort study analyzed 680 patients with ischemic stroke and 151 patients with hemorrhagic stroke at six hospitals between April 2015 and October 2021. All patients completed the functional status measures before rehabilitation (baseline), and at the 12th week and 2nd year after rehabilitation. Patients in the post-acute care (PAC) group exhibited significantly larger improvements for the functional status measures compared to those in the non-PAC group (*p* < 0.05). Patients with hemorrhagic stroke also displayed larger improvements in the functional status measures when compared to patients with ischemic stroke. Furthermore, the improvement in MCID ranged from 0.01 to 16.18 points when comparing baseline and the 12th week after rehabilitation, but the deterioration in MCID ranged from 0.38 to 16.12 points. Simultaneously, assessing the baseline and the second year after rehabilitation, the improvement in MCID ranged from 0.01 to 18.43 points, but the deterioration in MCID ranged from 0.68 to 17.26 points. Additionally, the PAC program, age, education level, body mass index, smoking, readmission within 30 days, baseline functional status score, use of Foley catheter and nasogastric tube, as well as a history of previous stroke are significantly associated with achieving deterioration in MCID (*p* < 0.05). These findings suggest that if the mean change scores of the functional status measures have reached the thresholds, the change scores can be perceived by patients as clinically important.

## 1. Introduction

It has been demonstrated that the post-acute care (PAC) program plays a crucial role in aiding stroke patients to return to their homes and communities by significantly improving functional status outcomes [1,2,3]. Utilizing patient-reported functional status measures has emerged as a valuable approach to examine functional recovery after stroke, gaining increased interest in recent years [4,5]. Functional status measures have now become standard evaluation tools for assessing stroke outcomes. Despite the growing recognition of the positive effects of PAC programs, previous studies have some notable limitations [6,7,8,9]. Firstly, only a few articles have utilized longitudinal data with more than two time points, and fewer still have explored functional status predictors over periods exceeding 12 weeks. Secondly, the number of studies focusing on PAC is increasing, but most studies analyzed either a small number of patients or solely those treated at one medical institution. Thirdly, longitudinal analyses in such studies often fail to employ appropriate statistical methodologies to account for censoring and intercorrelations arising from repeated measures within the same patient pool. Addressing these shortcomings is crucial for gaining deeper insights into the impact of PAC programs on stroke patient outcomes.

One common method used to assess outcomes after stroke is to determine the Minimum Clinically Important Difference (MCID) [10,11]. This represents the smallest change in outcome that a patient would perceive as beneficial. Given that clinical outcomes can vary from one patient to another, the MCID has garnered attention as a clinically meaningful threshold value above which the outcome is deemed relevant by the patient. While many studies have examined the MCID, there has been a notable absence of research evaluating the MCID after post-stroke acute care programs [12,13,14]. Enhancing our understanding of the MCID and identifying factors that predict deterioration in the MCID can significantly impact the definition of worse outcomes and the development of shared decision-making methods. As clinical relevance takes precedence over mere statistical significance, utilizing the MCID, rather than relying solely on patient-reported outcomes, is crucial in providing more meaningful insights into the efficacy of stroke rehabilitation programs.

As far as our knowledge extends, there are currently no studies that have established clinically significant changes in the scores of functional status measures after post-stroke acute care. Therefore, the main objective of this study was to assess the temporal trend, determine the MCID, and identify risk factors associated with achieving deterioration in the MCID of the functional status measures for patients with stroke.

## 2. Methods

### 2.1. The PAC-CVD Program

The Post-Acute Care—Cerebrovascular Diseases (PAC-CVD) program was initiated in Taiwan in 2014 with the aim of enhancing functional status outcomes for patients with stroke [3,4]. This program was meticulously designed, bringing together a multidisciplinary team comprising neurologists, physiatrists, physiotherapists, occupational therapists, speech therapists, and registered nurses. Prescribed by the physiatrist, the PAC program encompasses a comprehensive regimen of universal activities that are conducted at least three times per day. In contrast, the non-PAC program limits physical therapy, occupational therapy, and speech/swallowing therapy to once per day. Further elaboration on the specifics of the PAC-CVD program can be found in our previous works [3,4,5].

### 2.2. Study Population and Study Design

The inclusion criteria for this study were based on specific ICD-9-CM (430 and 431 for hemorrhagic stroke; 433.x, 434.x, and 436.x for ischemic stroke) and ICD-10 (I60–I62 for hemorrhagic stroke; I63 for ischemic stroke) codes for stroke types. Patients were considered eligible if they were admitted to a PAC ward at two medical centers or a non-PAC ward at three regional hospitals and one district hospital in south Taiwan between August 2015 and August 2021. Furthermore, patients needed to meet certain enrollment criteria, including being hospitalized for their first-ever stroke, having undergone computed tomography or magnetic resonance imaging examination within 30 days, and having a Modified Rankin Scale (MRS) score of 2 to 4. The MRS scale ranges from 0 to 5, where scores of 0, 1, 2, 3, 4, and 5 indicate no symptoms, no significant disability, slight disability, moderate disability, moderately severe disability, and severe disability, respectively [15]. Patients were excluded if PAC beds were unavailable at the participating hospitals or if they had been transferred to PAC wards at other hospitals. Initially, a total of 953 patients with ischemic stroke and 210 with hemorrhagic stroke were available during the study period. After excluding those who did not meet the inclusion criteria and those who declined to participate, a total of 795 ischemic stroke patients and 173 hemorrhagic stroke patients were initially recruited into the study (Figure 1 and Figure 2). The study protocol received approval from the institutional review board of Kaohsiung Medical University Hospital (KMUH-IRB-20140308), and informed consent was obtained from each participant before their enrollment in the study.

### 2.3. Instruments and Potential Covariates

In the present study, the measures of functional status were assessed using the Barthel Index (BI) [16], Functional Oral Intake Scale (FOIS) [17], Instrumental Activities of Daily Living Scale (IADL) [18], Berg Balance Scale (BBS) [19], and Mini-Mental State Examination (MMSE) [20]. All stroke patients completed the surveys before rehabilitation (baseline) and in the 12th week and 2nd year after rehabilitation. The Chinese versions of all instruments used in this study have been validated and used extensively in both clinical practice and research [3,4,5].

A research assistant performed questionnaire interviews at three different time points and reviewed medical records to obtain information related to stroke outcomes. The covariates of the chart review included age, gender, education, body mass index, smoking, drinking, average length of stay, readmission in 30 days, recurrence, score of functional status at baseline, Foley catheter, nasogastric tube, hypertension, diabetes mellitus, hyperlipidemia, atrial fibrillation, and previous stroke.

### 2.4. The Anchor Question

Using a global rating of change scale, one anchor question was used to assess the current functional status of the PAC-CVD program at the time of study completion. The first question was phrased, “compared to right before the PAC-CVD program, how would you describe the functional status now?” Response to the question was chosen from a 5-point Likert scale: much better, slightly better, no change, slightly worse, and much worse.

### 2.5. The Minimal Clinical Important Difference (MCID)

Wells et al. conducted a comprehensive literature review and classified MCID determination into nine different methods [21]. Similarly, Mouelhi and colleagues performed a systematic review and categorized these methods into anchor-based and distribution-based approaches [22]. For the current study, the MCID for the functional status measures was determined using the anchor-based method. To estimate the MCID in the global rating of change scale, this study utilized the global rating of change scale as the evaluation tool for functional status measures [23,24]. The study involved patients with stroke who underwent intensive rehabilitation programs, thereby assuming the potential for changes in functional status measures when estimating the MCID in the global rating of change scale. For the anchor-based approach, the study determined the change in the MRS from follow-up assessments (at 12 weeks and 2 years after rehabilitation) to the baseline assessment (before intensive rehabilitation programs). This change encompassed at least one level, including the achievement group, stable group, and deterioration group.

### 2.6. Statistical Analysis

The unit of analysis in this study was the individual patient with either ischemic stroke or hemorrhagic stroke. After assessing the distribution of observed patients and the number of patients excluded due to loss to follow-up, death, or refusal to participate at various time points, the baseline data for the study population were compared based on their participation in the PAC-CVD program. Continuous variables were evaluated for statistical significance using one-way analysis of variance (ANOVA), while categorical variables were analyzed using Fisher exact analysis. These comparisons were carried out to examine any significant differences between the two groups in terms of their baseline characteristics.

Using a propensity score approach, we applied inverse probability of treatment weighting (IPTW) analysis to balance the baseline study characteristics between the PAC group and non-PAC group. Each observation was weighted using the inverse of the probability of a patient receiving the PAC-CVD program, given observed potential covariates identified to the index date [25]. To ensure the stability of the inverse probability weights and reduce the impact of very small estimated probabilities, we used stabilized inverse probability weights derived from the propensity score model. The primary goal was to obtain estimates representing population average treatment effects while achieving optimal balance between these two groups. The treatment assignment was based on the method chosen at the time of study consent. Regression models were used for the final inference, which allowed for adjustments for potential covariates that remained unbalanced after IPTW [26].

Effect size (ES) was also calculated to directly compare the relative magnitude of change as measured via the two measures. That is, ES was calculated as the difference between the mean scores for two time intervals divided by the standard deviation of the score for the previous (or former) time interval [27]. Using this method of standardizing the extent of change assessed via one measure enabled comparisons between two measures. An ES of 1.0 is equivalent to a change of one standard deviation in the sample. Effect sizes of 0.2, 0.5, and 0.8 are typically considered small, medium, and large changes, respectively.

Univariate and multivariate logistic regression analyses were conducted to evaluate which potential covariates were significantly associated with the attainment of a deterioration in MCID for each functional status measure. Odds ratios and their 95% CIs were calculated for the functional status measures in respect to achieving the deterioration in MCID. Statistical analyses were conducted using SPSS software (Version 23.0; IBM). *p* < 0.05 was considered to indicate a statistically significant difference.

## 3. Results

### 3.1. Study Characteristics of Patients before and after IPTW

After excluding patients who were lost to follow-up or died, 680 patients with ischemic stroke and 151 patients with hemorrhagic stroke were finally recruited into the study (Table 1). After adjustment for IPTW, all potential covariates were well balanced.

### 3.2. Temporal Trends in Functional Status Measures at Different Time Points

Both patients with ischemic stroke and patients with hemorrhagic stroke in the PAC group had a larger absolute ES for all functional status measures compared to patients in the non-PAC group between baseline and the 12th week after rehabilitation (Table 2). In the same time period, both patients in the PAC group and in the non-PAC group showed the largest ES in the BBS simultaneously. In comparisons between baseline and the 12th week after rehabilitation, however, all patients with stroke showed a lower absolute ES for the functional status measures between the 2nd year after rehabilitation and the 12th week after rehabilitation, and during this time period, all patients with stroke showed the largest absolute ES in the BI simultaneously. Additionally, during the study period, patients with stroke in the PAC group showed a larger absolute ES for the functional status measures compared to patients with stoke in the non-PAC group, at the same time point, but patients with hemorrhagic stroke had larger absolute ES in most functional status measures than patients with ischemic stroke.

### 3.3. MCIDs in Functional Status Measures at Different Time Points

The mean change, standard deviation, and MCIDs of the functional status measures in patients with stroke at different time points after IPTW are shown in Table 3. In patients with ischemic stroke, between baseline and the 12th week after rehabilitation, the MCIDs in the functional status measures for “improvement” vs. “maintain (no change)” were from 0.66 to 16.18 in the PAC group and from 0.12 to 10.40 in the non-PAC group; in patients with hemorrhagic stroke, the MCIDs were from 0.01 to 9.25 in the PAC group and from 0.39 to 8.82 in the non-PAC group simultaneously. Similarly, in patients with ischemic stroke, between baseline and the second year after rehabilitation, the MCIDs were from 0.64 to 18.43 in the PAC group and from 0.37 to 9.40 in the non-PAC group; in patients with hemorrhagic stroke, the MCIDs were from 0.01 to 10.47 in the PAC group and from 0.06 to 7.96 in the non-PAC group simultaneously. In patients with ischemic stroke, between baseline and the 12th week after rehabilitation, the MCIDs for “deterioration” vs. maintain (no change)” were from 0.89 to 16.12 in the PAC group and from 0.81 to 11.44 in the non-PAC group; in patients with hemorrhagic stroke, the MCIDs were from 1.13 to 10.23 in the PAC group and from 0.38 to 12.10 in the non-PAC group simultaneously. Similarly, in patients with ischemic stroke, between baseline and the second year after rehabilitation, the MCIDs were from 0.83 to 17.26 in the PAC group and from 0.68 to 14.88 in the non-PAC group; in patients with hemorrhagic stroke, the MCIDs were from 1.11 to 10.81 in the PAC group and from 1.00 to 12.09 in the non-PAC group simultaneously.

### 3.4. Risk Factors of MCIDs in Functional Status Measures at Different Time Points

In patients with ischemic stroke, multivariate logistic analysis generally showed that PAC program, age, education, body mass index, smoking, readmission in 30 days, score of functional status at baseline, Foley catheter, nasogastric tube, and previous stroke were risk factors associated with deterioration in MCIDs in the functional status measures (Table 4). Similarly, in patients with hemorrhagic stroke, it also generally showed that PAC program, readmission in 30 days, score of functional status at baseline, Foley catheter, nasogastric tube, and previous stroke were risk factors.

## 4. Discussion

Patients in the PAC group exhibited significantly larger ES for functional status measures compared to those in the non-PAC group. Patients with hemorrhagic stroke also displayed larger ES when compared to patients with ischemic stroke. Furthermore, the improvement in MCID ranged from 0.01 to 16.18 points when comparing baseline and the 12th week after rehabilitation, but the deterioration in MCID ranged from 0.38 to 16.12 points. Simultaneously, assessing baseline and the second year after rehabilitation, the improvement in MCID ranged from 0.01 to 18.43 points, but the deterioration in MCID ranged from 0.68 to 17.26 points. Additionally, factors that were generally associated with achieving deterioration in MCID included participation in the PAC program, age, education level, body mass index, smoking, readmission within 30 days, baseline functional status score, use of a Foley catheter and nasogastric tube, as well as a history of previous stroke.

Observational studies inherently carry the risk of confounding due to both measurable and unmeasurable variables. To mitigate this concern, we adopted a pragmatic non-randomized cluster design with the use of IPTW. This strategic approach aimed to establish comparability in baseline characteristics, resembling the structure of a randomized clinical trial for our primary outcome. Importantly, this design not only augmented our sample size robustly but also alleviated the inherent biases associated with individual selection commonly encountered in observational studies based on registries. The incorporation of IPTW marks a notable methodological advancement, enabling us to rigorously align the PAC and non-PAC groups in terms of factors potentially associated with confounding variables. Consequently, our study’s findings reflect a sense of equilibrium in these comparisons, thereby bolstering our confidence in the adjusted estimates’ capacity to account for the presence of potential confounding variables. This encompasses variables that have been previously scrutinized in prior investigations as well as those meticulously integrated into our analytical framework. The comprehensive integration of these measures not only elevates the credibility of our study but also underscores its methodological rigor.

The MCID was implemented in this study to provide clinically significant outcomes that are perceived as a minimal benefit to stroke patients, rather than merely relying on statistically significant values [10,11,12]. The MCID offers not only a meaningful interpretation of intervention effects but also assists in obtaining more accurate sample sizes in research. Its importance has been highlighted in numerous studies, especially in the field of stroke, encompassing both ischemic and hemorrhagic cases [28,29]. It is worth noting that the MCID can vary depending on different diseases and rehabilitative programs. While previous studies have determined the MCID following inpatient stroke rehabilitation, there has been a lack of determination of the MCID after post-acute care, which represents a continuous and comprehensive rehabilitation program for stroke patients [23,30]. In this study, we sought to establish the MCID and assess the predictors of achieving deterioration in the MCID for the functional status measures among stroke patients.

Various methods have been utilized to establish the MCID. However, a definitive consensus on the best approach is yet to be reached. These methods can be broadly categorized into two groups: the distribution-based method and the anchor-based method [22,31]. The distribution-based method relies on statistical significance and lacks a direct link to clinical significance. It often employs the standard deviation as a statistical measure to determine the MCID. Other techniques include the standard error of measurement and minimal detectable change, which account for measurement errors in patient-reported outcomes, regardless of the patient population. On the other hand, the anchor-based method sets the MCID threshold using clinical or patient-based anchor questions. It encompasses various approaches, such as receiver operating characteristic (ROC) analysis, the mean change method, and the mean difference method. Among these, ROC analysis is the most commonly used method and is recommended by the Food and Drug Administration (FDA) as the most accessible approach to determining the MCID within an individual analysis. The mean change method calculates the MCID based on the absolute change in mean measure scores from baseline to follow-up in patients who respond with “somewhat better” to anchor questions. In contrast, the mean difference method derives the MCID by comparing measure scores between two transitional groups, such as “somewhat better” and “no change”. The mean difference method is particularly appropriate for comparing the study group and control group in a clinical trial. In this study, the anchor-based method was chosen to determine the patient-based MCID. By relying on individual subjective responses to assess improvement, this method provided valuable insights into patients’ perceptions of meaningful changes.

Following stroke rehabilitation, patients with hemorrhagic stroke showed higher functional status scores than those with ischemic stroke. This difference is likely due to the generally more severe nature of hemorrhagic stroke compared to ischemic stroke, as well as the lower functional status scores at baseline for patients with hemorrhagic stroke [32]. Interestingly, the group receiving PAC demonstrated higher functional status scores compared to the non-PAC group. Moreover, the differences in total scores for each functional status measure between the two groups increased from baseline to 12 weeks after rehabilitation, eventually reaching a plateau at two years after rehabilitation. This study, conducted with a Taiwanese population, revealed that the provision of essential rehabilitative care was more extensive in PAC patients compared to non-PAC patients. The differences in total scores for each functional status measure between the PAC and non-PAC groups gradually grew, indicating that the PAC program significantly enhanced standard follow-up care and potentially increased the utilization of available services. These findings align with similar results reported in previous studies [3,4,5].

Comparing the MCIDs in functional status measures with those reported in previous studies was not possible due to the lack of available research in this area. However, some studies have explored the MCID after stroke rehabilitation using the BBS. In one retrospective clinical analysis of 80 early subacute stroke patients, the estimated MCID of BBS scores was 5 points in the assisted-walking group and 4 points in the unassisted-walking group [33]. Another prospective assessment of 75 inpatients with acute stroke found the MCID in the BBS to be between 6.5 and 12.5 points, determined via the anchor-based approach [34]. Furthermore, these values were lower than the MCID values obtained in the present study. The differences in the type of stroke evaluated may account for the variations in the MCID results. Additionally, variations in analytical methods and patient factors, such as age, gender, disease severity, functional status at baseline, and timing of the evaluation, could also contribute to the observed differences between studies.

It is essential to not only calculate the MCID but also identify risk factors associated with achieving deterioration in the MCID. By pinpointing relevant factors, it becomes possible to make predictions and adjust post-rehabilitation outcomes through appropriate patient education for individuals with stroke. However, few studies have demonstrated a relationship between the potential covariates and deterioration in the MCID among patients with stroke. Notably, to the best of our knowledge, this is the first study to identify risk factors associated with achieving deterioration in the MCID in the functional status measures for patients with stroke. In our investigation, we discovered that the PAC program and functional status at baseline were factors associated with achieving deterioration in the MCID, consistent with findings from earlier studies that linked these factors to post-stroke rehabilitation quality of life and functional status [3,4,5]. Moreover, a higher functional status at baseline was found to decrease the odds ratios of experiencing deterioration in the MCID, in line with the results of other studies [35,36]. This suggests that patients with poorer functional status at baseline may have more room for clinically meaningful improvement, leading to higher potential for positive rehabilitation outcomes.

This study boasts several notable strengths. Notably, it is the first population-based prospective cohort study designed to assess the temporal trend, establish the MCID, and identify factors associated with deterioration in the MCID in the functional status measures for patients with stroke. Furthermore, comprehensive and highly reliable patient characteristic data were obtained through a certification process from related medical charts, which is strictly regulated by the Bureau of National Health Insurance (BNHI). This ensures the reliability and accuracy of the study population’s data. Additionally, we employed the IPTW method, which allowed us to create comparison groups that were well balanced in all baseline characteristics. This rigorous statistical approach enhances the validity and robustness of the study’s findings.

This study has several limitations that should be acknowledged. Firstly, the follow-up period was relatively short, spanning only 2 years. To provide a more comprehensive understanding of the MCID for treatment outcomes, future studies should consider evaluating it over an extended follow-up period. Secondly, the MCID was calculated using the anchor-based method, relying on the subjective judgment of patients with stroke. While this approach offers valuable patient-centric insights, incorporating objective measures in future research could complement the findings. Thirdly, the MCID after stroke rehabilitation, as reported in our study, was based on data from multiple centers, potentially limiting the generalizability of our findings. However, it is worth noting that the distribution of patients in our study was similar to that observed in previous studies [2,37], suggesting a reasonable level of generalizability. Fourthly, the assessment of acute ischemic stroke severity often employs the National Institutes of Health Stroke Scale (NIHSS). Nevertheless, within the scope of this study, the MRS score was adopted as the primary outcome measure in the context of stroke clinical trials. Fifthly, the clinical utility of mechanical thrombectomy in cases of posterior circulation large vessel occlusion (pc-LVO) remains uncertain when compared to its application in the anterior circulation [38,39]. Notably, the present study does not differentiate between the anterior and posterior circulation. Finally, our study may be subject to selection bias, as it only included patients who completed the functional status measures. Despite these limitations, this study is pioneering in determining the MCID and investigating factors related to deterioration in the MCID.

## 5. Conclusions

In conclusion, early and intensive rehabilitative PAC following stroke has shown promising results in enhancing functional restoration. The study observed a significant improvement in total scores for each functional status measure, with notable progress from baseline to 12 weeks after rehabilitation, eventually stabilizing at the two-year follow-up. Moreover, this study determined the MCID thresholds for the functional status measures at the two-year follow-up. The findings revealed that participation in the PAC program, baseline functional status score, and patient characteristics played predictive roles in the deterioration in MCIDs. The study findings suggest that reaching specific thresholds in the mean change scores of functional status measures is indicative of clinically important changes for patients. This insight holds promise for designing comprehensive care programs to optimize outcomes for stroke patients.

## Figures and Tables

**Figure 1 jcm-12-05828-f001:**
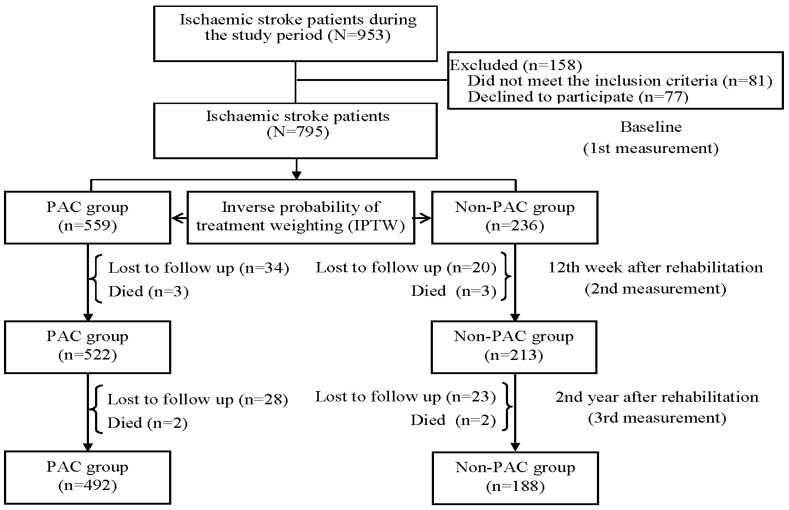
Flowchart of the study population. Patients with ischemic stroke (*n* = 680).

**Figure 2 jcm-12-05828-f002:**
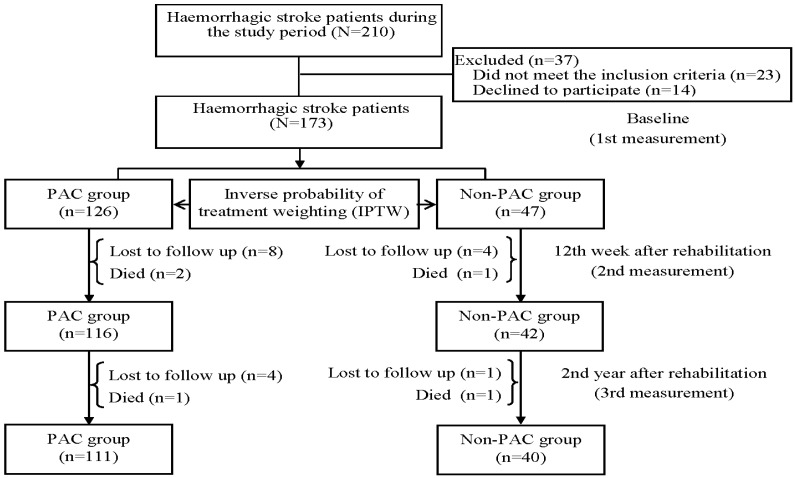
Flowchart of the study population. Patients with hemorrhagic stroke (*n* = 151).

**Table 1 jcm-12-05828-t001:** Study characteristics of patients with stroke before and after inverse probability of treatment weighting (IPTW) ^†^.

Variables	Before IPTW			After IPT		
	PAC Group	Non-PAC Group	*p* Value	PAC Group	Non-PAC Group	*p* Value
**Ischaemic stroke (*n* = 680)**						
Age, years	69.91 ± 12.42	71.43 ± 13.25	0.162	70.48 ± 14.38	72.4 ± 25.61	0.890
Gender, male (%)	314 (63.8%)	117 (62.2%)	0.768	64.3%	60.9%	0.220
Education, years	7.93 ± 1.90	8.37 ± 4.91	0.097	7.9 ± 2.41	7.78 ± 9.24	0.730
Body mass index, kg/m^2^	24.08 ± 3.72	24.34 ± 3.38	0.389	24.31 ± 4.52	24.27 ± 6.39	0.890
Smoking, yes (%)	84 (17.1%)	63 (33.5%)	<0.001	23.3%	20.6%	0.250
Drinking, yes (%)	40 (8.1%)	44 (23.4%)	<0.001	11.9%	12.0%	1.000
Average lengths of stay, days	18.66 ± 9.42	14.96 ± 9.35	<0.001	17.19 ± 11.1	17.87 ± 22.25	0.480
Readmission in 30 days, yes (%)	23 (4.7%)	16(8.5%)	0.082	8.0%	6.0%	0.940
Recurrence, yes (%)	6 (1.2%)	25 (13.3%)	<0.001	6.0%	4.9%	0.460
Modified Rankin Scale, MRS = 2 (%)	17 (3.5%)	10 (5.3%)	2.429	5.8%	4.9%	0.790
MRS = 3 (%)	64 (13.0%)	30 (16.0%)		12.5%	12.6%	
MRS = 4 (%)	411 (83.5%)	148 (78.7%)		81.7%	82.5%	
Foley catheter, yes (%)	41 (8.3%)	34(18.1%)	<0.001	10.4%	12.5%	0.220
Nasogastric tube, yes (%)	120 (24.4%)	54 (28.7%)	0.289	24.8%	28.9%	0.090
Hypertension, yes (%)	352 (71.5%)	134 (71.3%)	1.000	70.3%	69.8%	0.880
Diabetes mellitus, yes (%)	222 (45.1%)	71 (37.8%)	0.100	46.1%	45.3%	0.060
Hyperlipidemia, yes (%)	218 (44.3%)	55 (29.3%)	<0.001	38.4%	37.4%	0.690
Atrial fibrillation, yes (%)	59 (12.0%)	18 (9.6%)	0.451	11.2%	13.8%	0.170
Previous stroke, yes (%)	80 (16.3%)	47 (25.0%)	0.012	19.7%	22.2%	0.290
**Hemorrhagic stroke (*n* = 151)**						
Age, years	62.19 ± 12.57	68.9 ± 14.63	0.006	62.1 ± 14.31	65.94 ± 27.81	0.140
Gender, male (%)	69 (62.2%)	27 (67.5%)	0.682	60.06%	55.4%	0.490
Education, years	8.22 ± 1.36	7.45 ± 5.62	0.398	8.28 ± 1.84	7.96 ± 5.65	0.560
BMI, kg/m^2^	23.86 ± 4.64	23.77 ± 3.2	0.901	23.77 ± 5.23	23.04 ± 5.99	0.250
Smoking, yes (%)	12 (10.8%)	9 (22.5%)	0.118	15.0%	14.6%	1.000
Drinking, yes (%)	5 (4.5%)	10 (25.0%)	0.001	9.4%	11.3%	0.670
Average lengths of stay, days	22.21 ± 8.14	15.38 ± 12	<0.001	22.08 ± 9.7	23.04 ± 19.78	0.600
Readmission in 30 days, yes (%)	4 (3.6%)	7 (17.5%)	0.008	19.0%	23.0%	0.150
Recurrence, yes (%)	4 (3.6%)	3 (7.5%)	0.382	3.7%	3.3%	1.000
Modified Rankin Scale, MRS = 2 (%)	1 (0.9%)	1 (2.5%)	0.733	1.1%	1.1%	0.880
MRS = 3 (%)	10 (9.0%)	4 (10.0%)		10.0%	11.7%	
MRS = 4 (%)	100 (90.1%)	35 (87.5%)		88.9%	87.2%	
Foley catheter, yes (%)	13 (11.7%)	8 (20.0%)	0.302	13.4%	13.4%	1.000
Nasogastric tube, yes (%)	31 (27.9%)	9 (22.5%)	0.647	25.8%	21.6%	0.460
Hypertension, yes (%)	101 (91.0%)	29 (72.5%)	0.009	83.6%	81.7%	0.770
Diabetes mellitus, yes (%)	28 (25.2%)	15 (37.5%)	0.204	25.1%	20.5%	0.410
Hyperlipidemia, yes (%)	18 (16.2%)	11 (27.5%)	0.187	20.9%	23.5%	0.650
Atrial fibrillation, yes (%)	2 (1.8%)	3 (7.5%)	0.116	3.3%	3.4%	1.000
Previous stroke, yes (%)	9 (8.1%)	4 (10.0%)	0.746	8.5%	7.1%	0.830

^†^ Values are expressed as mean ± standard deviation or *n* (%). Abbreviation: MRS, modified rankin scale.

**Table 2 jcm-12-05828-t002:** Temporal trends in the functional status measures at different time points during the study period after IPTW.

	PAC Group							Non-PAC Group						
Measure	Baseline (T1)	12th Week after Rehabilitation (T2)			2nd Year after Rehabilitation (T3)			Baseline (T1)	12th Week after Rehabilitation (T2)			2nd Year after Rehabilitation (T3)		
	Mean ± SD	Mean ± SD	ES ^†^	*P* ^†^	Mean ± SD	ES ^‡^	*P* ^‡^	Mean ± SD	Mean ± SD	ES ^†^	*p* ^†^	Mean ± SD	ES ^‡^	*p* ^‡^
**Ischaemic stroke (*n* = 680)**														
BI	39.18 ± 27.17	58.37 ± 27.50	0.33	<0.001	59.58 ± 27.40	0.04	0.044	40.11 ± 27.18	56.75 ± 26.52	0.30	<0.001	57.60 ± 32.63	0.03	0.040
FOIS	4.90 ± 2.38	6.00 ± 1.54	0.26	<0.001	6.10 ± 1.76	0.06	0.139	5.35 ± 2.42	5.78 ± 3.38	0.01	0.800	5.83 ± 3.64	0.01	0.649
IADL	0.87 ± 0.52	2.00 ± 1.20	0.45	<0.001	2.09 ± 1.98	0.08	0.089	1.28 ± 1.76	1.50 ± 1.43	0.24	0.488	1.60 ± 2.21	0.07	0.084
BBS	16.97 ± 19.80	30.94 ± 16.22	0.67	<0.001	32.02 ± 20.20	0.07	0.551	18.84 ± 13.91	22.99 ± 21.20	0.19	<0.001	23.47 ± 20.02	0.02	0.093
MMSE	18.06 ± 10.34	20.37 ± 8.80	0.14	<0.001	21.40 ± 10.10	0.12	0.251	18.99 ± 12.48	19.54 ± 14.30	−0.03	0.319	19.93 ± 14.04	0.03	0.184
**Haemorrhagic stroke (*n* = 151)**														
BI	29.88 ± 20.56	43.21 ± 26.24	0.75	<0.001	53.78 ± 26.56	0.40	<0.001	32.8 ± 21.93	47.04 ± 14.60	0.42	0.007	52.11 ± 17.70	0.35	<0.001
FOIS	5.43 ± 1.00	6.05 ± 1.05	0.46	<0.001	6.46 ± 0.95	0.39	0.031	5.36 ± 2.53	5.85 ± 1.90	0.17	0.059	5.93 ± 1.77	0.04	0.315
IADL	0.93 ± 0.65	1.30 ± 1.37	0.87	<0.001	1.61 ± 1.37	0.23	0.038	0.52 ± 1.20	0.89 ± 1.83	0.27	0.015	0.92 ± 1.90	0.02	0.296
BBS	13.12 ± 7.65	26.15 ± 12.02	1.48	<0.001	30.42 ± 12.61	0.36	<0.001	7.97 ± 6.34	13.74 ± 7.73	1.08	0.014	14.40 ± 8.92	0.05	0.062
MMSE	18.82 ± 8.06	22.06 ± 8.75	0.41	<0.001	22.70 ± 8.64	0.07	0.206	15.82 ± 12.64	17.47 ± 13.59	0.12	0.010	17.94 ± 12.77	0.03	0.082

^†^ T2 vs. T1, ^‡^ T3 vs. T2. Abbreviations: IPTW, inverse probability of treatment weighting; PAC, post-acute care; SD, standard deviation; ES, effective size; BI: Barthel Index; FOIS: Functional Oral Intake Scale; IADL: Instrumental Activities of Daily Living Scale; BBS: Berg Balance Scale; MMSE: Mini-Mental State Examination.

**Table 3 jcm-12-05828-t003:** Mean change, standard deviation (SD), and minimal clinically important difference (MCID) of the functional status measures in patients with stroke at different time points after inverse probability of treatment weighting (IPTW).

	PAC Group						Non-PAC Group					
Measure	12th Week after Rehabilitation vs. Baseline			2nd Year after Rehabilitation vs. Baseline			12th Week after Rehabilitation vs. Baseline			2nd Year after Rehabilitation vs. Baseline		
	Mean Change	SD	MCID	Mean Change	SD	MCID	Mean Change	SD	MCID	Mean Change	SD	MCID
**Ischaemic stroke (*n* = 680)**												
Barthel Index												
Improvement	29.79	17.02	16.18	31.84	16.79	18.43	21.13	17.34	10.40	22.94	17.43	9.40
Maintain (no change)	13.61	12.64		13.41	12.72		10.73	8.12		13.54	10.60	
Deterioration	−2.51	10.74	16.12	−3.85	1.74	17.26	−0.71	1.93	11.44	−1.34	1.73	14.88
Functional Oral Intake Scale												
Improvement	0.76	1.51	0.66	0.90	1.54	0.64	0.95	2.50	0.12	1.07	2.11	0.37
Maintenance (no change)	0.10	2.21		0.26	1.85		0.83	2.89		0.70	1.70	
Deterioration	−1.91	3.47	2.01	−1.16	3.46	1.93	0.02	1.47	0.81	0.02	1.97	0.68
Instrumental Activities of Daily Living Scale												
Improvement	1.42	1.21	0.98	1.36	1.21	0.92	1.95	1.74	1.63	2.07	1.66	1.79
Maintenance (no change)	0.44	0.83		0.44	0.85		0.32	1.08		0.28	1.23	
Deterioration	−0.45	0.53	0.89	−0.39	0.54	0.83	−0.81	0.88	1.13	−1.09	0.85	1.37
Berg Balance Scale												
Improvement	19.90	14.14	9.76	19.24	14.02	9.14	10.05	8.15	3.56	12.27	7.52	4.39
Maintenance (no change)	10.14	9.11		10.10	9.66		6.49	5.48		7.88	5.46	
Deterioration	−1.64	2.53	11.78	−3.44	1.81	13.54	−0.05	1.28	6.54	−0.21	1.78	8.09
Mini-Mental State Examination												
Improvement	2.59	1.23	1.19	3.62	2.21	1.11	1.56	5.55	1.42	1.67	5.37	2.31
Maintenance (no change)	1.40	0.40		2.51	1.81		0.14	1.09		−0.64	1.89	
Deterioration	−3.56	2.66	4.95	−3.61	2.83	6.12	−6.06	5.38	6.20	−3.08	3.52	2.44
**Haemorrhagic stroke (*n* = 151)**												
Barthel Index												
Improvement	14.75	8.79	5.02	15.79	9.65	5.58	14.05	6.78	5.16	14.00	5.47	4.51
Maintenance (no change)	9.73	4.58		10.21	5.37		8.89	4.02		9.49	5.73	
Deterioration	−0.50	1.00	10.23	−0.60	1.60	10.81	−3.21	2.11	12.10	−2.60	1.43	12.09
Functional Oral Intake Scale												
Improvement	1.03	1.87	0.01	1.05	1.91	0.01	0.76	1.14	0.39	0.55	0.61	0.06
Maintenance (no change)	1.04	1.53		1.04	1.52		0.37	1.18		0.49	1.45	
Deterioration	−1.75	0.50	2.79	−2.12	2.06	3.16	−0.29	0.76	0.66	−3.64	1.15	4.13
Instrumental Activities of Daily Living Scale												
Improvement	1.45	1.24	1.07	1.47	1.27	1.08	0.81	1.03	0.57	0.85	1.04	0.61
Maintenance (no change)	0.38	0.64		0.39	0.64		0.24	0.63		0.24	0.61	
Deterioration	−0.75	0.50	1.13	−0.72	0.52	1.11	−0.14	0.38	0.38	−0.76	0.90	1.00
Berg Balance Scale												
Improvement	13.48	5.17	9.25	15.68	6.60	10.47	15.07	7.50	8.82	14.40	10.82	7.96
Maintenance (no change)	4.23	0.64		5.21	3.09		6.25	3.33		6.44	4.65	
Deterioration	−2.50	3.00	6.73	−2.92	1.67	8.13	−1.71	4.54	7.96	−3.28	3.20	9.72
Mini-Mental State Examination												
Improvement	4.98	2.17	1.50	4.95	2.30	1.43	1.95	2.66	0.48	1.65	2.32	0.57
Maintenance (no change)	3.48	2.08		3.52	2.07		1.47	2.20		1.08	2.84	
Deterioration	−6.75	4.50	10.23	−5.68	3.09	9.20	−5.29	3.40	6.76	−8.04	1.34	9.12

**Table 4 jcm-12-05828-t004:** Factors associated with deterioration in minimal clinically importance difference in the functional status measures in patients with stroke at different time points.

	Ischaemic Stroke (*n* = 680)		Haemorrhagic Stroke (*n* = 151)	
Measure	Odds Ratio (95%CI)	*p* Value	Odds Ratio (95%CI)	*p* Value
Barthel Index (BI)				
12th week after rehabilitation vs. baseline				
Diabetes mellitus, yes	1.97 (1.04~3.73)	<0.001	–	
Post-acute care program, yes	0.96 (0.95~0.98)	<0.001	0.95 (0.91~0.99)	<0.001
Readmission in 30 days, yes	1.09 (1.04~1.14)	<0.001	–	
Nasogastric tube, yes	–		1.02 (1.01~1.04)	<0.001
Functional status at baseline, scores	0.98 (0.98~0.99)	<0.001	0.96 (0.94~0.98)	<0.001
2nd year after rehabilitation vs. baseline				
Post-acute care program, yes	0.97 (0.95~0.99)	<0.001	0.93 (0.89~0.97)	<0.001
Readmission in 30 days, yes	1.10 (1.08~1.12)	<0.001	–	
Nasogastric tube, yes	1.87 (1.67~2.01)	<0.001	1.48 (1.15~1.82)	<0.001
Functional status at baseline, scores	0.96 (0.95~0.97)	<0.001	0.96 (0.94~0.98)	<0.001
Functional Oral Intake Scale (FOIS)				
12th week after rehabilitation vs. baseline				
Body mass index, kg/m^2^	1.06 (1.01~1.12)	0.017	–	
Drinking, yes	3.53 (1.60~7.80)	0.001	–	
Post-acute care program, yes	0.94 (0.92~0.96)	<0.001	0.95 (0.90~0.99)	<0.001
Readmission in 30 days, yes	1.80 (1.08~3.00)	<0.001	–	
Recurrence, yes	2.76 (1.07~7.12)	<0.001	–	
Nasogastric tube, yes	2.15 (1.11~4.14)	<0.001	4.96 (1.65~6.93)	<0.001
Functional status at baseline, scores	0.44 (0.38~0.51)	<0.001	0.42 (0.28~0.63)	<0.001
2nd year after rehabilitation vs. baseline				
Body mass index, kg/m^2^	1.17 (1.10~1.25)	0.009	–	
Drinking, yes	3.91 (1.45~10.57)	<0.001	1.16 (1.09~1.24)	<0.001
Post-acute care program, yes	0.94 (0.90~0.98)	<0.001	0.92 (0.84~0.99)	<0.001
Nasogastric tube, yes	2.54 (1.59~4.06)	<0.001	1.09 (0.01~2.73)	<0.001
Hyperlipidemia, yes	1.84 (1.07~3.16)	<0.001	–	
Functional status at baseline, scores	0.34 (0.28~0.42)	<0.001	0.39 (0.23~0.67)	<0.001
Instrumental Activities of Daily Living Scale (IADL)				
12th week after rehabilitation vs. baseline				
Age, years	0.98 (0.97~0.99)	0.009	–	
Post-acute care program, yes	0.90 (0.87~0.92)	<0.001	0.87 (0.74~0.58)	<0.001
Nasogastric tube, yes	2.06 (1.26~3.36)	<0.001	4.36 (1.67~6.43)	<0.001
Atrial fibrillation, yes	1.84 (1.07~3.16)	<0.001	2.79 (1.03~7.55)	<0.001
Previous stroke, yes	2.13 (1.05~4.33)	<0.001	–	
Functional status at baseline, scores	0.78 (0.65~0.94)	<0.001	0.54 (0.38~0.76)	<0.001
2nd year after rehabilitation vs. baseline				
Education, years	1.14 (1.06~1.22)	<0.001	–	
Post-acute care program, yes	0.95 (0.92~0.99)	<0.001	0.92 (0.88~0.96)	<0.001
Foley catheter, yes	–		3.35 (0.83~0.93)	<0.001
Nasogastric tube, yes	2.37 (1.08~5.17)	<0.001	4.36 (1.67~6.43)	<0.001
Hyperlipidemia, yes	2.38 (1.30~4.37)	<0.001	4.79 (0.08~0.40)	<0.001
Functional status at baseline, scores	0.75 (0.61~0.93)	<0.001	0.51 (0.28~0.93)	<0.001
Berg Balance Scale (BBS)				
12th week after rehabilitation vs. baseline				
Age, years	0.98 (0.97~0.99)	0.031	–	
Education, years	1.09 (1.03~1.14)	<0.001	–	
Smoking, yes	–		2.77 (1.70~4.53)	<0.001
Post-acute care program, yes	0.94 (0.92~0.96)	<0.001	0.93 (0.89~0.97)	<0.001
Nasogastric tube, yes	2.94 (1.77~4.87)	<0.001	2.90 (1.21~6.92)	<0.001
Hypertension, yes	–		3.63 (1.36~9.73)	<0.001
Atrial fibrillation, yes	3.25 (1.36~7.20)	<0.001	–	
Previous stroke, yes	2.19 (1.22~3.92)	<0.001	2.86 (1.04~7.91)	<0.001
Functional status at baseline, scores	1.03 (1.01~1.04)	<0.001	0.81 (0.67~0.99)	<0.001
2nd year after rehabilitation vs. baseline				
Smoking, yes	6.27 (2.18~8.76)	<0.001	–	
Post-acute care program, yes	0.95 (0.93~0.97)	<0.001	0.91 (0.84~0.98)	<0.001
Hypertension, yes	1.71 (1.20~2.42)	<0.001	1.61 (1.17~2.23)	<0.001
Previous stroke, yes	7.12 (2.04~7.95)	<0.001	8.14 (3.65~7.68)	<0.001
Functional status at baseline, scores	0.94 (0.93~0.95)	<0.001	0.57 (0.40~0.81)	<0.001
Mini-Mental State Examination (MMSE)				
12th week after rehabilitation vs. baseline				
Education, years	1.12 (1.02~1.22)	0.001	1.19 (1.04~1.16)	<0.001
Readmission in 30 days, yes	–		5.29 (1.25~6.38)	<0.001
Foley catheter, yes	2.13 (1.05~4.33)	<0.001	–	
Previous stroke, yes	3.25 (1.36~7.20)	<0.001	3.91 (1.09~9.10)	<0.001
Functional status at baseline, scores	0.89 (0.85~0.92)	<0.001	0.95 (0.91~0.99)	<0.001
2nd year after rehabilitation vs. baseline				
Smoking, yes	6.10 (1.55~6.75)	<0.001	4.87 (1.07~6.69)	<0.001
Post-acute care program, yes	–		0.72 (0.53~0.96)	<0.001
Foley catheter, yes	3.54 (1.13~11.08)	<0.001	3.03 (1.40~5.57)	<0.001
Functional status at baseline, scores	0.76 (0.68~0.85)	<0.001	0.81 (0.73~0.91)	<0.001

## Data Availability

The data that support the findings of this study are available from the corresponding author upon reasonable request.

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
