# Peer review of "Minimal Clinically Important Difference (MCID) in the Functional Status Measures in Patients with Stroke: Inverse Probability Treatment Weighting"

_jcm, 2023, doi:10.3390/jcm12185828_

Round 1

Reviewer 1 Report

In the article authors present MCID as functional status measures at the two-year follow up in patients after stroke. They performed a prospective cohort study which included 680 patients with ischemic stroke and 151 patients with hemorrhagic stroke. Patients were from 6 hospitals, and data were collected around 6 years. Check-up points for functional status were at the beginning - baseline, then after 12th week and after 2 years. In the introduction they describe the backgrounds. Methods describe their PAC-CVD program, study population and study design, used instruments and potential covariates, what was "The Anchor Question", MCID and used statistical tools. The results section clearly present results divided in study characteristics of patients before and after IPTW, temporal trends of functional status measures at different time points, MCIDs of functional status measures at different time points and risk factors of MCIDs of functional status measures at different time points. Discussion section describes results, clearly compares and explains results, critically sees the limitations of study and in the end sum up the content in conclusion. Authors present that their PAC program, baseline functional status score and patient characteristics have predictive roles in the deterioration of MCID. Better results in the PAC program group are expected, while the program is more intense and performed in two medical centers. This is also limitation that should be pointed out. There is also no original or new information about factors that could contribute to deterioration.

Authors should structure the abstract. It would be useful, if they could more clearly point out how MCID of Functional Status Measures in Patients with Stroke help to evaluate stroke patients comparing with all already used scores, measures etc.

Author Response

Reviewer 1

In the article authors present MCID as functional status measures at the two-year follow up in patients after stroke. They performed a prospective cohort study which included 680 patients with ischemic stroke and 151 patients with hemorrhagic stroke. Patients were from 6 hospitals, and data were collected around 6 years. Check-up points for functional status were at the beginning - baseline, then after 12th week and after 2 years. In the introduction they describe the backgrounds. Methods describe their PAC-CVD program, study population and study design, used instruments and potential covariates, what was "The Anchor Question", MCID and used statistical tools. The results section clearly present results divided in study characteristics of patients before and after IPTW, temporal trends of functional status measures at different time points, MCIDs of functional status measures at different time points and risk factors of MCIDs of functional status measures at different time points. Discussion section describes results, clearly compares and explains results, critically sees the limitations of study and in the end sum up the content in conclusion. Authors present that their PAC program, baseline functional status score and patient characteristics have predictive roles in the deterioration of MCID. Better results in the PAC program group are expected, while the program is more intense and performed in two medical centers. This is also limitation that should be pointed out. There is also no original or new information about factors that could contribute to deterioration.

Ans: Thank you for your positive feedback. Building upon the insights you provided, we have further enriched the study by incorporating an additional paragraph that elucidates the implementation of the Inverse Probability of Treatment Weighting (IPTW) method. This method has proven to be an instrumental tool in addressing confounding variables and enhancing the robustness of our findings (lines 281-296).

Moreover, guided by your valuable input, we have carefully refined the statements in the Discussion section. Through this meticulous revision process, we aimed to ensure that the scientific discourse accurately reflects the methodological advancements and the comprehensive approach undertaken in our study. This iterative process, informed by your constructive comments, has undoubtedly strengthened the scholarly value of our work. (lines 359-374).

----------------------------------------------------------------------------------------

Authors should structure the abstract. It would be useful, if they could more clearly point out how MCID of Functional Status Measures in Patients with Stroke help to evaluate stroke patients comparing with all already used scores, measures etc.

Ans: Regardful of your comment above, the conclusions within the Abstract (lines 53-54) as well as the Conclusions section (lines 414-417) have been revised. Again, thank you very much.

Reviewer 2 Report

Thanks for the opportunity. 

The paper is well written and the topic is very actual, because of the importance of rehabilitation's pathway in ischemic and hemorrhagic stroke.

Interesting paper about early and intensive rehabilitative post-acute care. This study observed an improvement in many scores for each functional status measure and the participation in the PAC program predict the deterioration of MCIDs. 

But there is a big lack: there is no evaluation of the NIHSS for ischemic stroke.

You have to evaluate it, distinguishing anterior and posterior circulation. I can suggest you some papers about it: 

-https://doi.org/10.3390/life11121423

-https://doi.org/10.1136/jnis-2022-019557

Good English

Author Response

Thanks for the opportunity.

The paper is well written and the topic is very actual, because of the importance of rehabilitation's pathway in ischemic and hemorrhagic stroke.

Interesting paper about early and intensive rehabilitative post-acute care. This study observed an improvement in many scores for each functional status measure and the participation in the PAC program predict the deterioration of MCIDs.

Ans: Thank you for your encouraged comments above.

----------------------------------------------------------------------------------------

But there is a big lack: there is no evaluation of the NIHSS for ischemic stroke.

Ans: Following your comment above, we added this issue in the revised subsection of the Limitations (lines 395-399). Again, thank you very much.

----------------------------------------------------------------------------------------

You have to evaluate it, distinguishing anterior and posterior circulation. I can suggest you some papers about it:

-https://doi.org/10.3390/life11121423

-https://doi.org/10.1136/jnis-2022-019557

Ans: Following your advice above, we have incorporated the provided statements and references into the Limitations subsection, enhancing the scholarly depth of our study (lines 399-402). Again, thank you very much.

Round 2

Reviewer 2 Report

After the revision, the paper is improved with a good, general quality.